# Lightweight Design of Front Suspension Upright of Electric Formula Car Based on Topology Optimization Method

**Jixiong Li \* , Jianliang Tan and Jianbin Dong**

Foshan University, Foshan 528000, China; fstjl@fosu.edu.cn (J.T.); djb666jbd@gmail.com (J.D.)
\* Correspondence: Lijx@fosu.edu.cn

**Abstract:** In order to obtain a lightweight front upright of an electric formula car's suspension, the topology optimization method is used in the front upright structure design. The mathematical model of the lightweight optimization design is constructed, and the geometric model of the initial design of the front upright is subjected to the ultimate load condition. The structural optimization of a front upright resulted in the mass reduction of the upright by 60.43%. The optimized model was simulated and verified regarding the strength, stiffness, and safety factor under three different conditions, namely turning braking, emergency braking, and sharp turning. In the experiment, the uprights were machined and assembled and integrated into the racing suspension. The experimental results showed that the optimized front uprights met the requirements of performance.

**Keywords:** front suspension upright; topology optimization method; lightweight design; physical experiment

## 1. Introduction

The front upright is one of the key components of a racing suspension assembly, which connects the suspension link control arm, tie rod, and axle. The main function of the front upright is to transmit all the forces and moments between the wheel and the connecting rod control arm [1]. The suspension is an essential assembly to ensure the ride comfort and steering stability of a car, as well as the ideal motion characteristics of wheels when a road or load changes. In addition, to provide sufficient strength, the stiffness and durability need to meet the dynamic and static evaluation requirements of the Formula Racing competition [2]. Moreover, it is necessary to reduce the mass of the front upright structure as much as possible to improve the energy-saving performance of a car.

In recent years, many studies on the lightweight design of components of both Chinese and the foreign formula racing cars have been conducted. Li Fang [3] used a variable density topology optimization method to design a light racing frame. Yuan Shouli [4] used the SIMP topology optimization method to lightweight the racing frame. Hou Zhanfeng [5] proposed the topology optimization method for the design of a light racing rocker arm. Li Renren [6] used a topology optimization method to ensure light weight of the racing car bracket. At present, the lightweight design of racing parts is mainly concentrated on the frame and brackets. There are a few aspects related to the lightweight design and structural optimization of the front suspension of the racing suspension. By analyzing static and dynamic characteristics of the front upright of the racing suspension, this paper proposes a variable-density topology optimization method to lighten the design of uprights according to the requirements for competition conditions and structural design.

## 2. Topology Optimization Method

Topology optimization represents one of the three types of structural optimization. This kind of optimization is mainly used to obtain the optimal layout of structural materials. Accordingly, it has been widely used in the structural lightweight design of various products [7]. The design optimization of the assembly parts of the automotive products is to determine which portion of the parts can be removed to reduce the weight while keeping all the parts safe and reliable during the load transfer process [8].

At present, the topology optimization method of a product's structure includes the variable thickness method, homogenization method, variable density method, and others. Among them, the variable density method has been widely used in structural optimization design. In this method, the element material is used as a density, and discrete variables change between 0 and 1. Among them, the empty material of distribution is defined as 0, and the complete material of distribution is defined as 1, which can be considered as an optimization of the combination of 0 and 1 discrete variables. Then, the deletion and retention of a set of elements are determined based on certain criteria after many times of iteration optimization, where the structural units that are large for the structure transfer force path are retained, and units that are not useful are removed. Essentially, the structural topology optimization problem based on the variable density method denotes a discrete optimization problem, involving the increase and decrease of the cell. The corresponding mathematical model [9] can be expressed as follows:

$$Find\ \rho = \{\rho_1, \rho_2, \ldots, \rho_n\}^T \tag{1}$$

$$Min\ C(\rho) = F^T U \tag{2}$$

$$S.t\ V(\rho)/V_0 \leq f \tag{3}$$

$$KU = F \tag{4}$$

$$\rho_i = \{\mathbf{0, 1}\}\,(i = \mathbf{1, 2}, \ldots n) \tag{5}$$

where $C(\rho)$ denotes the flexibility of a structure, the objective function indicates the stiffness of the structure, $\rho_i$ denotes the design variable of an element and relative density of material, $F$ denotes the load matrix, $U$ denotes the displacement matrix, $K$ denotes the overall stiffness matrix; $V(\rho)$ represent the structural volume under the design variable state, where $V_0$ is the initial volume of the structure, and $f$ is the volume constraint parameter.

In view of this, the specific mathematical model of the front upright optimization design of the racing suspension can be expressed as follows:

$$Design\ variable : X = dev(com1) \tag{6}$$

$$Response\ type :\ h(x) = dis \tag{7}$$

$$g(x) = Vol \tag{8}$$

$$Restrictions :\ g(x) \leq N\% \tag{9}$$

$$Objective\ function :\ f(x) = \min(h(x)) \tag{10}$$

In Equations (6)–(10), *dev* denotes the cell mesh density change, *com*1 denotes the designable region, *dis* repressnts the static displacement, *vol* is the volume fraction which is equal to the difference between the total volume in the current iteration step and the ratio of the initial non-design area volume to the initial design area volume; constraints for the upper limit of the mass fraction, that is, the retained material, cannot exceed the material of the selected space, the objective function minimizes the displacement [10].

## 3. Topology Optimization of Front Upright of Racing Suspension

*3.1. Front Upright Topology Optimization Process*

The front upright of a racing car represents a component that connects the connecting rod control arm and the axle. Along with the functional design of the connecting hole, other non-connected area structures are considered in the lightweight design based on the structural characteristics and light weight of the front upright. In the topology optimization design, first, the topology optimization design analysis is conducted, as shown in Figure 1.

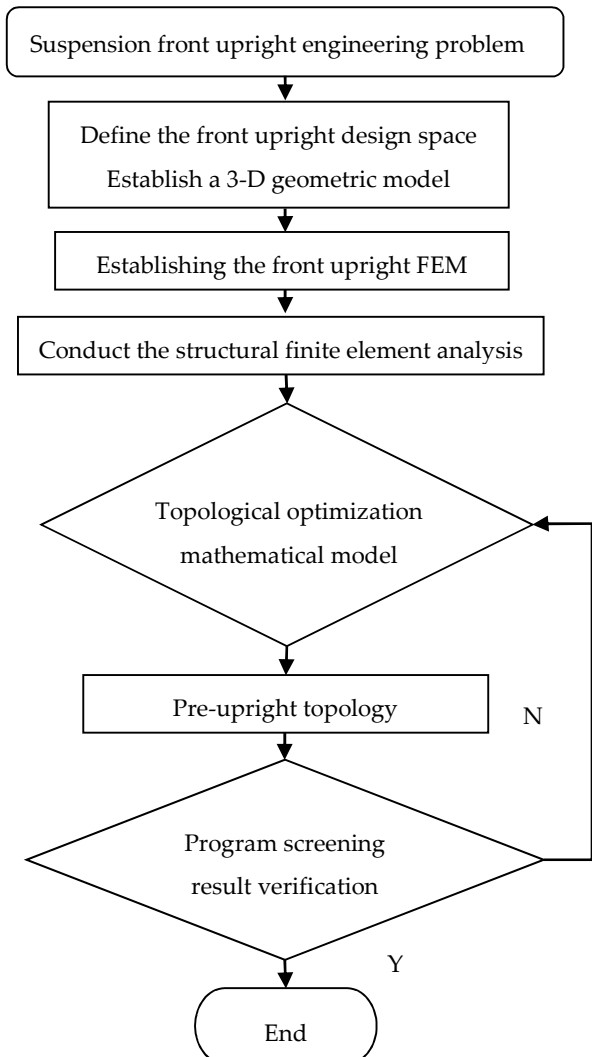

**Figure 1.** Topology optimization process for suspension front upright.

In the optimization design analysis of the front upright of the suspension, the preliminary overall size of the front upright is determined based on the structure and installation space of the suspension system of a car, and then, the hole of the connecting area, such as the connecting rod of the connecting rod, is determined according to the functional requirements of the front upright. The basic dimensions are used to establish a geometric model of the initial topology optimization, as shown in the first two blocks in Figure 1. Next, a finite element analysis of the structural strength characteristics of the established front upright geometry model is performed to provide a basis for optimizing the area setting. Then, the parameters of the topology optimization equation, including the variables, constraints, and objective functions of the front upright structure optimization, are set. In the next

step, the topology optimization and result analysis verification are conducted. If the optimized front upright structure solution does not meet the requirements of performance, then the parameter and topology of the structure need to be re-calculated. These processes are repeated until the predefined requirements are met.

*3.2. Determination of Initial Geometric Space and Establishment of Analytical Model*

The preliminary front upright geometry is determined based on the formula front wheel hub, inner and outer bearings, the size of the inner wall of the front upright, the mounting point coordinates of the upper suspension upper and lower control arms, the position of the steering tie rod, and the structural characteristics of the front upright design, as shown in Figure 2. According to the rules of the competition and the driving conditions of the racing car, it is known that the front upright of the suspension is subjected to the maximum load under the condition of turning braking, and thus, the topology optimization of the initial front upright model is conducted under that working condition.

First, the established geometric model is meshed, where the grid unit size of the main force area, such as holes, is 0.5 mm, and of the other areas is 1 mm; and the inner and outer bearing contact surfaces of the upright are layered by the grid control method; then according to the bending and braking conditions imposed on the upright and the load, as shown in Figure 3, constraining the six degrees of freedom of the inner wall of the intermediate connection hole, in the upper and lower control arm mounting hole, steering tie rod outer bearing hole, system. The moving caliper-mounting hole and the inner and outer hub-bearing holes are precisely loaded according to the coordinates of the hard point. The magnitude and direction of each action load are also shown in Figure 3. The finite element analysis model (FEM) is shown in Figure 3. The boundary conditions and loads for FEM are derived from suspension analysis calculations.

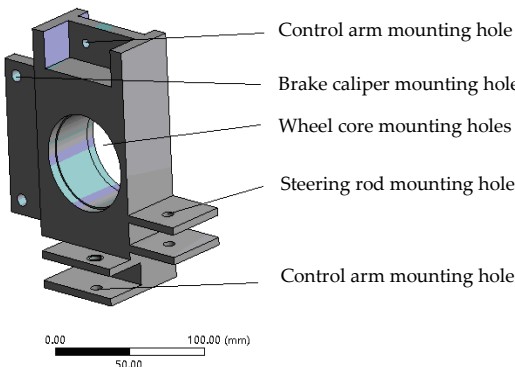

**Figure 2.** The initial geometric model.

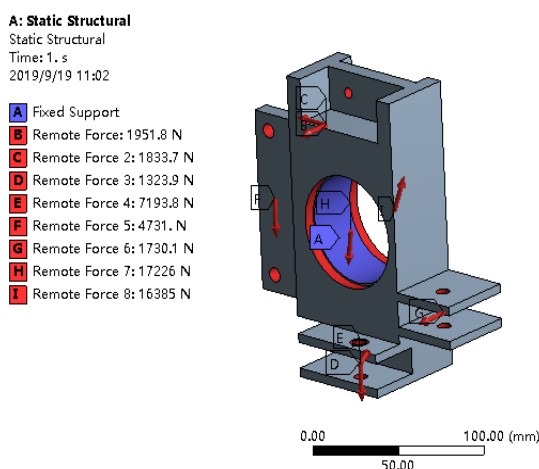

**Figure 3.** The initial finite element model (FEM).

### 3.3. Strength Analysis of Initial Model

The initial finite element model was calculated and analyzed, and the stress and deformation contours of the formula front upright under static loading that are respectively shown in Figures 4 and 5, were obtained. The maximum stress value obtained from the contour diagram was 132.49 MPa, and the maximum deformation was 0.17 mm. The 7075 aluminum material was used, and its yield strength is 455 MPa. Therefore, the material properties of the initial structure of the front upright of the designed racing suspension were not fully utilized, and the weight reduction can be further optimized.

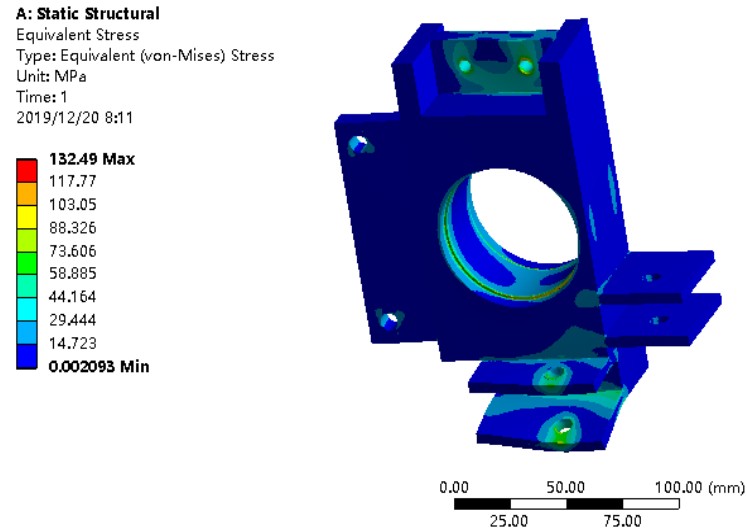

**Figure 4.** Stress contour.

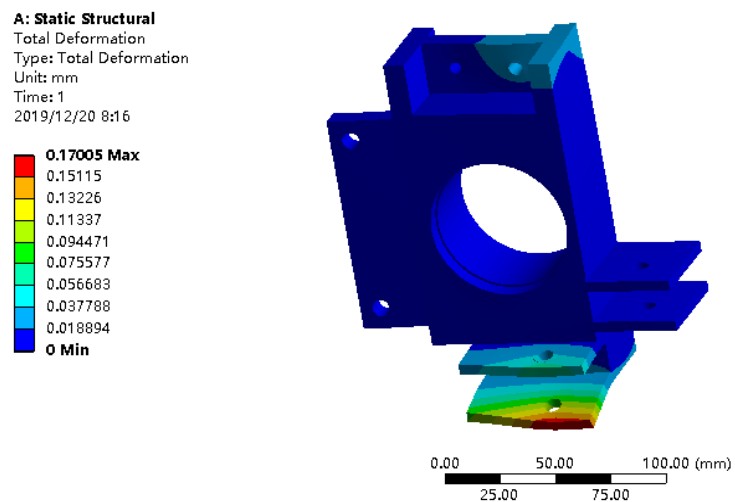

**Figure 5.** Deformation contour.

### 3.4. Topological Optimization of Front Upright

The initial design mass of the racing suspension front upright is 1.2056 kg, which is optimized for lightweight topology. According to the results of the aforementioned static analysis, the topological range is selected as shown in Figure 6, where the blue area denotes the optimized area while the red area denotes the non-optimized area, and the optimization constraint is set to 60% of the selected area mass. According to the set topology optimization three-factor parameters for optimization, the structure of the front upright optimization is obtained, and it is shown in Figure 7. The structure that is presented in Figure 7 is then transformed into a three-dimensional (3D) geometric model. By using the

3D geometric model, contour post-processing is performed, and the work includes the repair of curved lines and face merging. The processed structure is required to be as close as possible to the actual topological results, the surface is smooth, the structure is regular, and it is conducive to mechanical manufacturing processing. Finally, the geometrical space structure of the upright before the initial topology optimization is obtained, and it is shown in Figure 8. The model mass of this structure is 0.61438 kg, so the mass is reduced by 0.59122 kg.

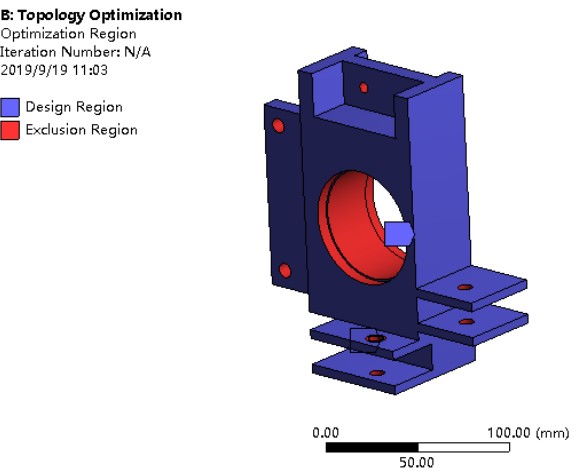

**Figure 6.** Topology optimization area.

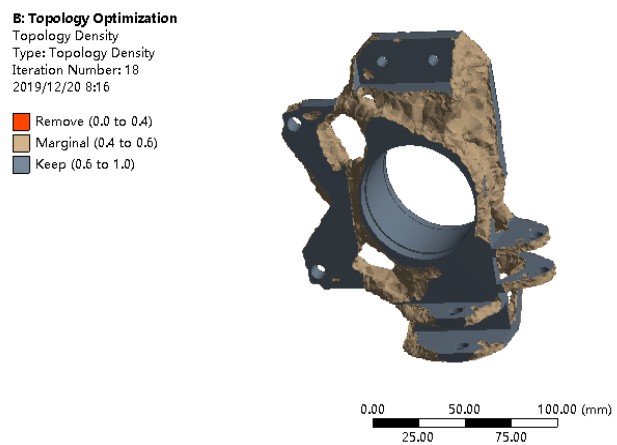

**Figure 7.** Topology optimization result.

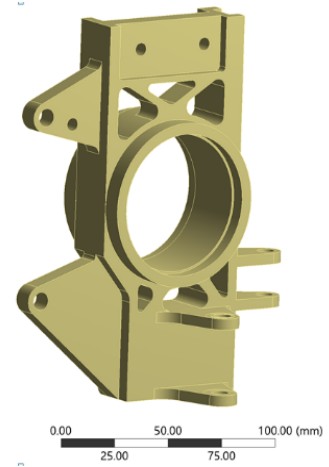

**Figure 8.** Geometric model after initial optimization.

After the initial topology optimization, static analysis is performed on the geometric model, wherein the meshing principle and imposed constraints and loads are consistent with the initial analysis. The calculation results are shown in Figures 9 and 10. In Figure 9, the maximum stress value is 227.47 MPa, while the maximum deformation value in Figure 10 is 0.41 mm; the safety factor is 2.22, both within the safety range of the aluminum material used, and the maximum stress value is much smaller than the yield limit, so the optimized structure weight is further reduced. The comparative analysis of the static analysis results of the contour image shows that the brake caliper installation area, steering rod installation area, and left and right installation areas of the hub bearing have certain optimization space. In view of this, further optimization of the hollowing out and weight reduction of these areas is conducted. The final mass of the suspension front upright geometry structure is 0.4771 kg, and it is shown in Figure 11.

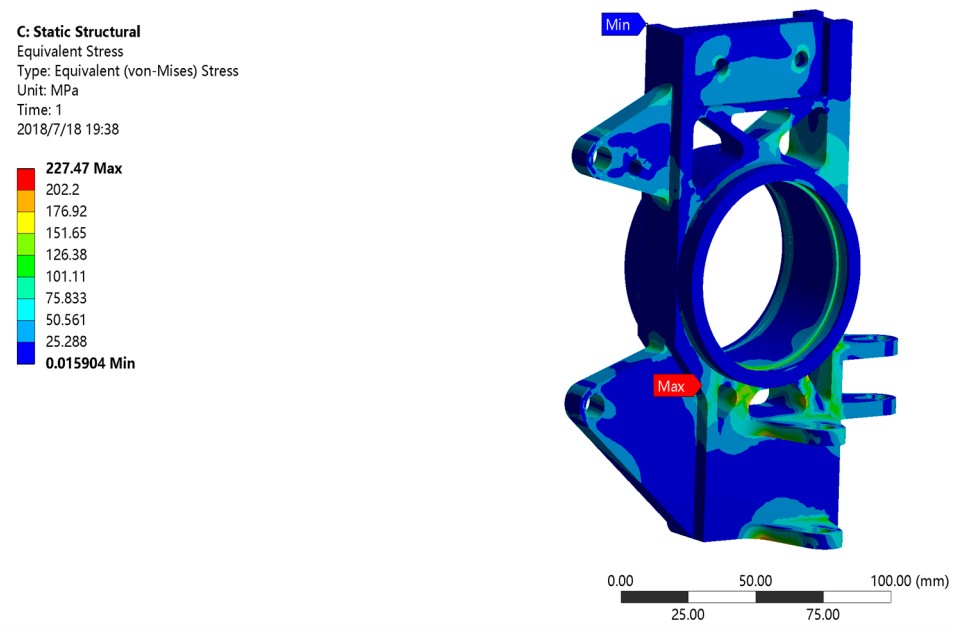

**Figure 9.** Stress contour.

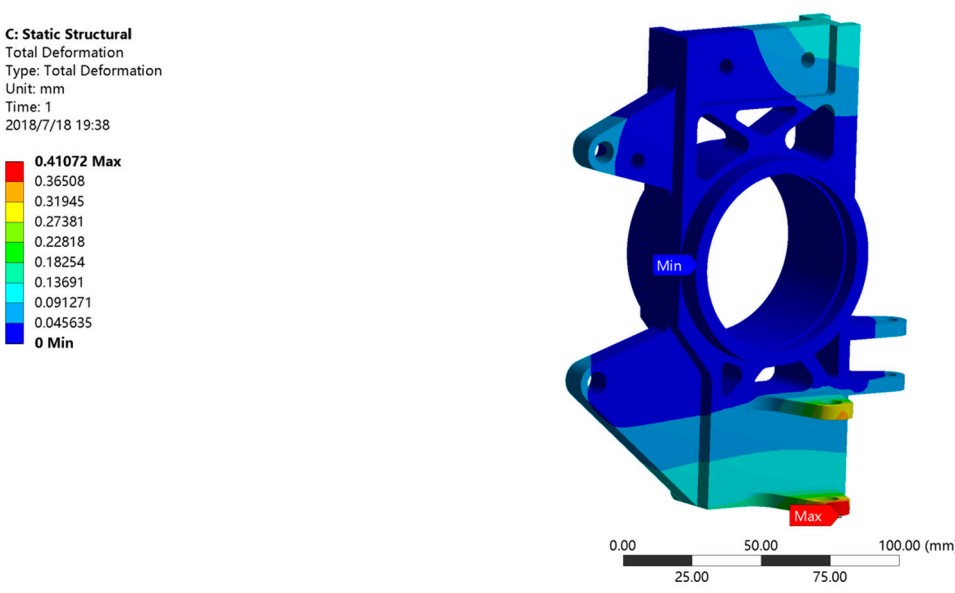

**Figure 10.** Deformation contour.

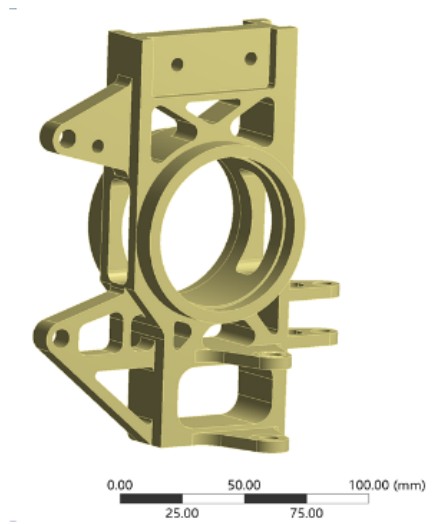

**Figure 11.** Final geometric model.

### 3.5. Optimization Results Analysis and Simulation Verification

3.5.1. Analysis and Verification under Turning Brake Conditions

In the final optimized geometric model, the meshing, constraints, and load application settings, and the aluminum alloy material property parameter settings, in which the meshing principle is consistent with the initial condition, and the constraint boundary conditions are consistent with the initial static analysis conditions, and each load at the joint is consistent with the initial analysis conditions according to the turning brake condition. The stress contour and deformation contour are calculated, and they are shown in Figures 12 and 13, respectively. The maximum stress value in Figure 12 is 211.71 MPa, and the maximum deformation value in Figure 13 is 0.58. The safety factor is 2.38, which meets the requirements for safety performance.

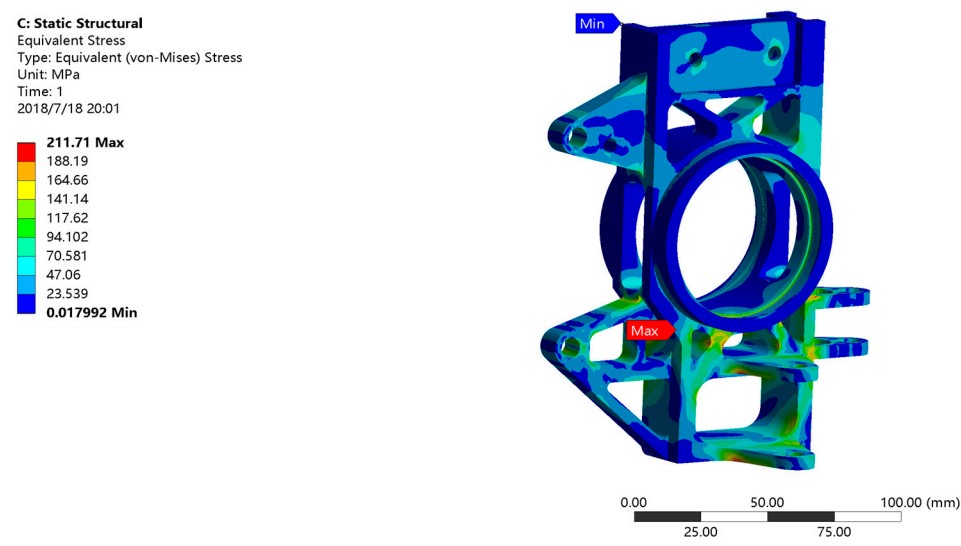

**Figure 12.** Stress contour.

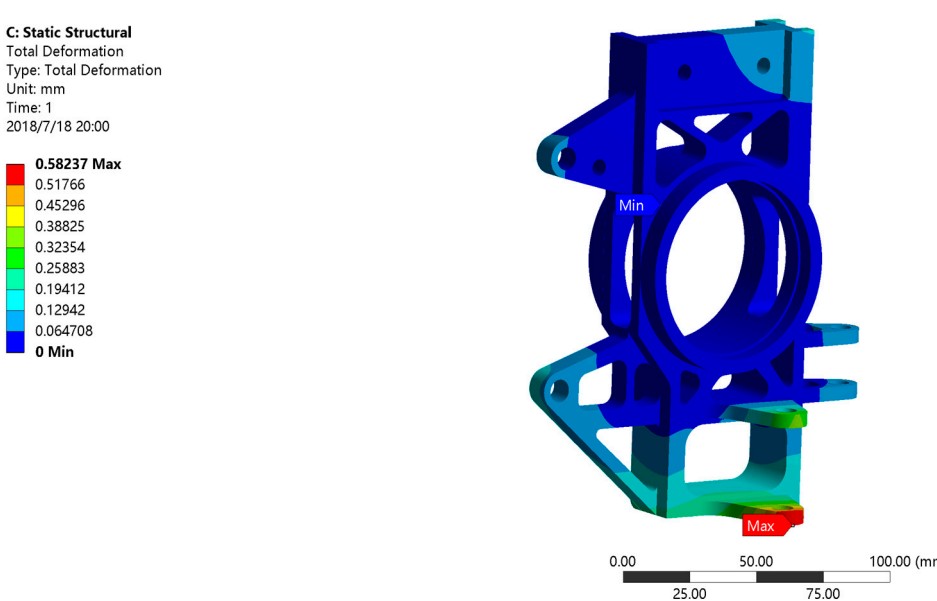

**Figure 13.** Deformation contour.

### 3.5.2. Analysis and Verification under Emergency Braking Conditions

Similar to the analysis of the sharp turn condition, the geometric model is separately defined by meshing, material parameter setting, constraint boundary conditions, and load conditions. The load applied to each connecting hole is calculated according to the emergency braking condition, and the suspension is established. The finite element analysis model of the front upright is shown in Figure 14. The stress contour and deformation contour obtained by the analysis and calculation are shown in Figures 15 and 16, respectively. The maximum stress value in Figure 15 is 223.MPa, and the maximum deformation value in Figure 16 is 0.393. The safety factor is 2.26, which meets the safety performance requirements.

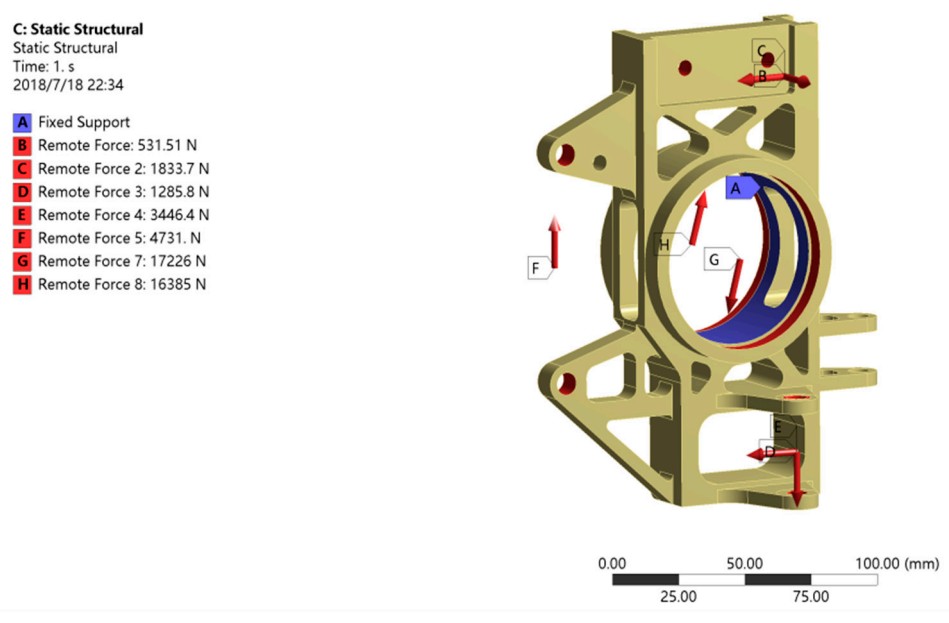

**Figure 14.** Finite element model.

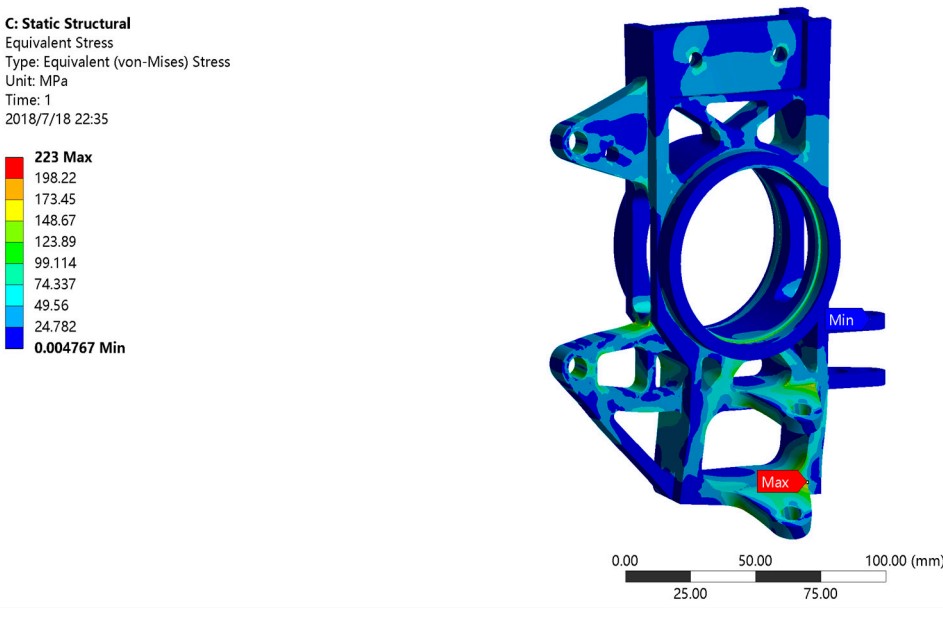

**Figure 15.** Stress contour.

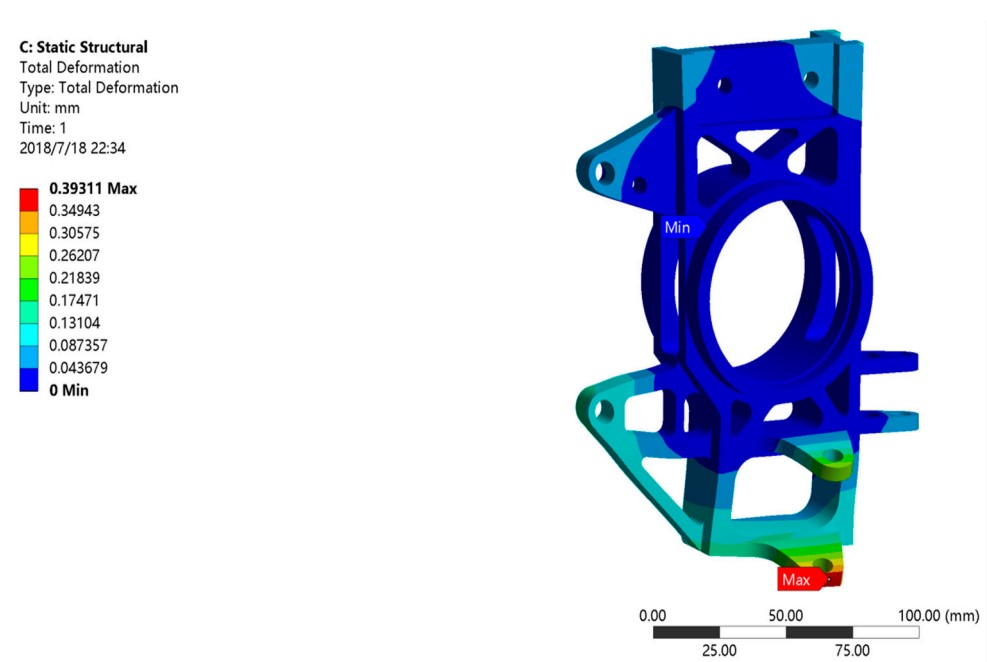

**Figure 16.** Deformation contour.

### 3.5.3. Analysis and Verification under Emergency Turning Conditions

Similar to the analysis of the sharp turn condition, the geometric model is separately defined by meshing, the material parameter setting, constraint boundary conditions, and load conditions. The load applied at each connecting hole is calculated according to the emergency turning condition, and the suspension is established. The front upright finite element analysis model obtained is shown in Figure 17. The stress contour and deformation contour obtained by the analysis and calculation are shown in Figures 18 and 19, respectively. The maximum stress value in Figure 18 is 199.57 MPa, and the maximum deformation value in Figure 19 is 0.5586. The safety factor is 2.53, which meets the requirements for safety performance.

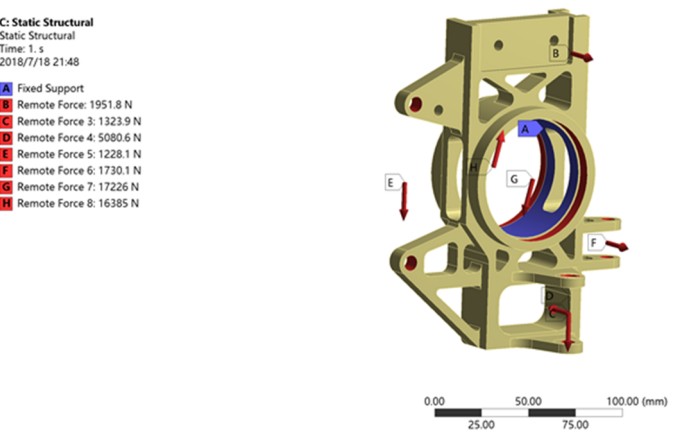

**Figure 17.** Finite element model.

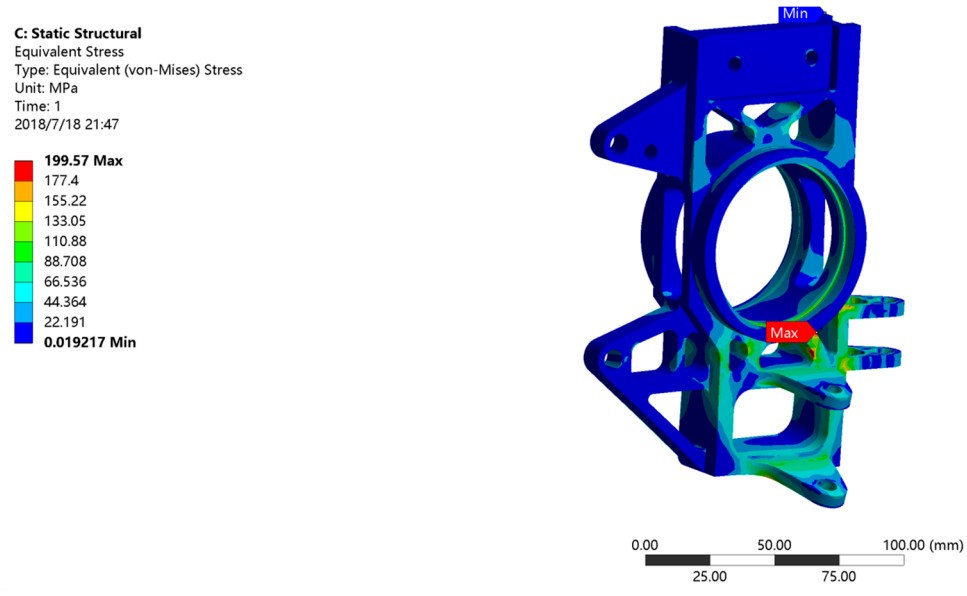

**Figure 18.** Stress contour.

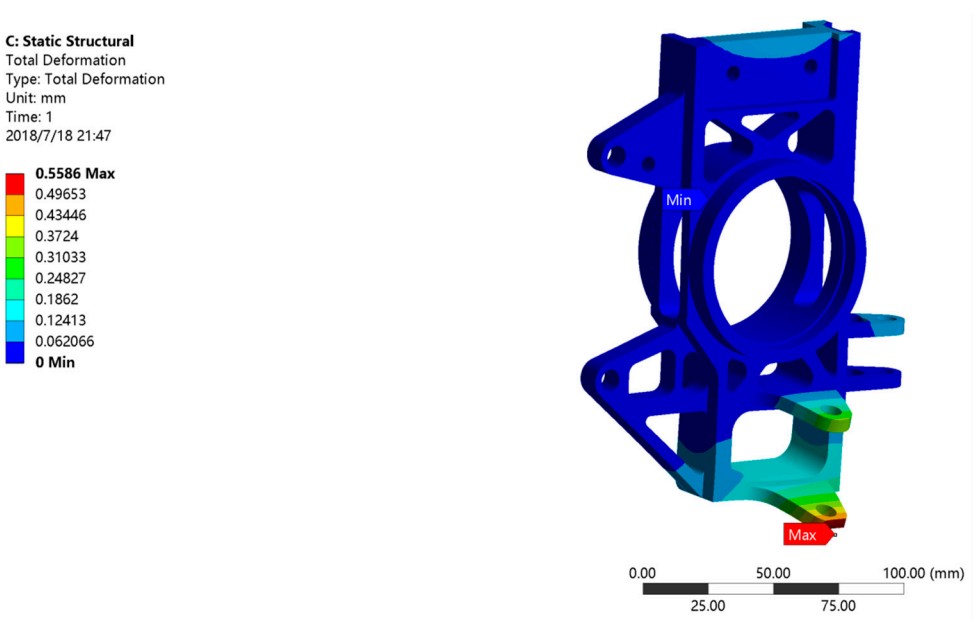

**Figure 19.** Deformation contour.

### 4. Experiment Verification of Front Upright

According to the structural characteristics of the topologically optimized geometric model obtained by the finite element analysis of the suspension front upright to meet the performance requirements, the mechanical manufacturing process was designed. The design process steps were as follows: blanking, rough grinding, fitter scribing, milling shape, drilling, wire cutting, and grinding deburring [11]. It was processed according to the design process steps to obtain the physical parts, as shown in Figure 20.

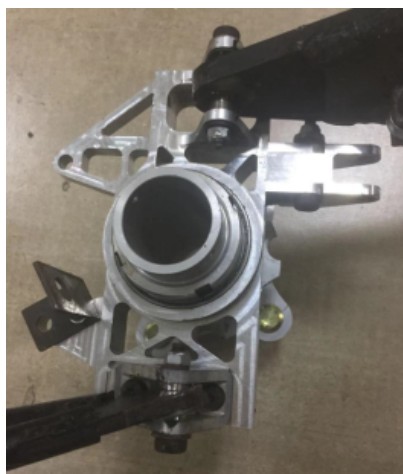

**Figure 20.** Manufactured part.

Finally, the front upright was integrated into the car, and the car test is shown in Figure 21. The vehicle was tested on the campus road, and the basic accelerated turning and deceleration braking performance tests were performed; no abnormality was found in the front upright of the suspension. In November 2018, we officially participated in the evaluation of the static project and dynamic project of the Chinese Formula Student in Zhuhai. The static projects mainly included the marketing report, racing design, cost, and manufacturing analysis, while dynamic projects included the linear acceleration test, figure of eight winding test, high-speed obstacle avoidance test, efficiency test, and a 22-km endurance test. The tested vehicle is presented in Figure 22. During the evaluation process, the front upright did not appear abnormal. After the evaluation was completed, it was transported back to the school car laboratory, and then the key components of the car suspension were dismantled. Figure 23 shows the disassembled front upright, where it can be seen that no obvious deformation or cracking failure phenomenon was presented, which indicated that the optimized topology design was light enough. The suspension front upright was quantified to meet the basic requirements of performance.

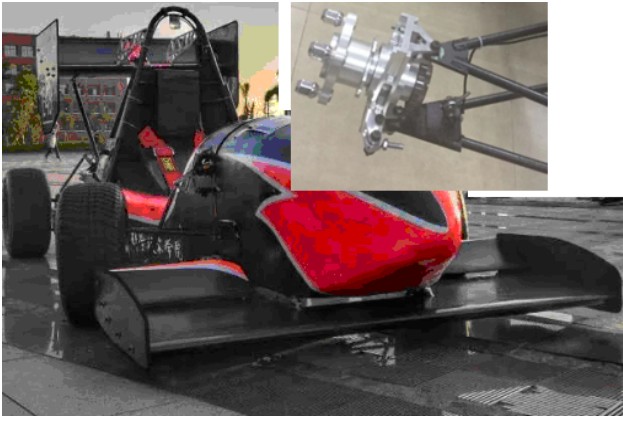

**Figure 21.** Assembly of integrated front uprights.

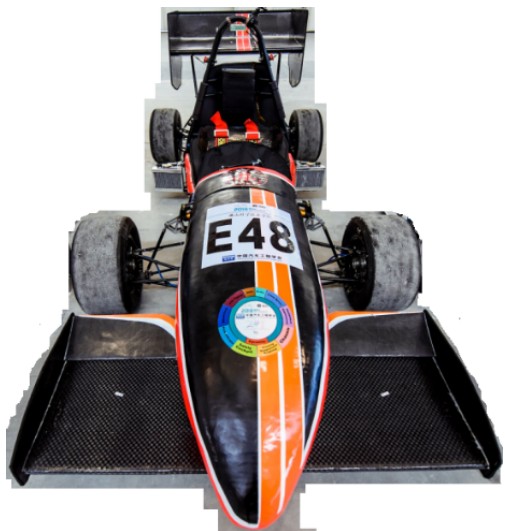

**Figure 22.** Tested vehicle.

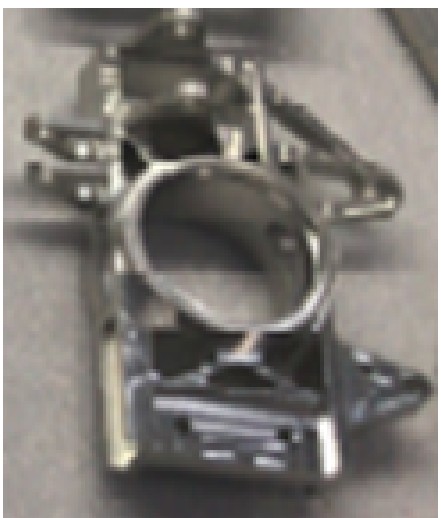

**Figure 23.** The front upright disassembled after the test.

## 5. Conclusions

The main contributions of this work can be summarized as follows.

(1)   By analyzing the structural topology optimization method, the mathematical model for the topology optimization of the front suspension of the racing suspension is established, and the initial geometry of the front upright is designed according to the suspension requirements provided by the 2018 Chinese University Formula Competition.

(2)   Then, using the established topology optimization mathematical model, the initial geometry model of the suspension front upright is optimized for the purpose of weight reduction, and the lightweight front upright with the mass reduction of 60.43% is obtained; the turning brake condition, emergency braking, and emergency turning are adopted. The strength, stiffness, and safety factor of the front upright are simulated under three extreme conditions. The results show that the lightweight front upright of the virtual design meets the requirements of the work.

(3)   The processing route of the front upright racing suspension is designed, the physical prototype is manufactured, and the assembly is integrated into a racing car. An experiment using the vehicle is conducted. The experimental result shows that the designed and manufactured lightweight suspension front upright meets the basic requirements of performance.

**Author Contributions:** Formal analysis, J.L. and J.D.; Validation, J.T.; Writing—original draft, J.L.; Writing—review and editing, J.L. All authors have read and agreed to the published version of the manuscript.

**Funding:** This paper is supported by the construction project of the electric vehicle scientific research basic innovation platform of Foshan Science and Technology Bureau of Guangdong Province (No. 2014AG10012).

**Conflicts of Interest:** The authors declare no conflict of interest.

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
