# Peer review of "Lightweight Design of Front Suspension Upright of Electric Formula Car Based on Topology Optimization Method"

_wevj, doi:10.3390/wevj11010015_

Round 1
Reviewer 1 Report
This is good work on optimizing the structural design for lightweighting and structural performance. And that the optimized shape has been manufactured and proven to be reliable based on testing results. Many sentences need be rewritten as they are too long and vague, making them hard to understand. Below are several examples from this paper:
Finally, the simulated effect and strength of the strength, stiffness and safety factor under three conditions of turning braking, emergency braking and sharp turning are applied to the optimized model.
Since the suspension is an important assembly to ensure the ride comfort and steering stability of the car, as well as the ideal motion characteristics of the wheel when the road changes and the load changes, in addition to the need to have sufficient strength, Stiffness and durability to meet the dynamic and static evaluation requirements of the Formula Racing competition [2], also need to have lightweight features to improve the energy-saving effect of the car.
Specific to the design of the assembly parts of the automotive products, that is, which parts of the parts can be removed to reduce the weight, and the optimized structure of the parts is safe and reliable during the load transfer process [8].
The variation between discrete variables, where the material of the distribution is defined as 0, and the material of the distribution is completely defined as 1, which can be understood as the optimization of the combination of 0-1 discrete variables, and then determine the set of a unit according to certain criteria.
In the process of optimization design analysis of the front pillar of the suspension, the preliminary overall size of the front pillar is determined according to the structure and installation space of the suspension system of the car, and then the hole of the connecting area such as the connecting rod of the connecting rod is determined according to the functional requirements of the front pillar.
The figure titles are not clear. For example, Figure 4, instead of “stress cloud”, it would be better explained if “stress cloud/contour of the formula front wheel hub under static loading” were used.
The quality of the initial model of the front suspension of the racing suspension is 1205.6g. Why the unit of “quality” is g? It should be mass, right? For mass of vehicle components, kg is used more often instead of g.
Does the mesh size have much influence on the FEM results? In Figure 7, some area has very large mesh size, while the rest area has much smaller mesh. Please explain.
Author Response
Point 1:The figure titles are not clear. For example, Figure 4, instead of “stress cloud”, it would be better explained if “stress cloud/contour of the formula front wheel hub under static loading” were used.
Response 1: add explanation “and the stress cloud and deformation cloud of the formula front wheel hub under static loading shown in Figures 4 and 5 “ in 2.3.Strength analysis of the initial model
Point 2: The quality of the initial model of the front suspension of the racing suspension is 1205.6g. Why the unit of “quality” is g? It should be mass, right? For mass of vehicle components, kg is used more often instead of g.
Response 2 :Yes, “kg” is used
Point 3:Does the mesh size have much influence on the FEM results? In Figure 7, some area has very large mesh size, while the rest area has much smaller mesh. Please explain.
Response 3: Yes, The mesh size have much influence on the Fem results, This is first formula car, This year we will design the second formula car. We will use smaller mesh.
Thanks!
Reviewer 2 Report
The subject of the present paper is interesting and significant and falls within the subject area of the journal. The topology optimization procedure presented in Figure 1 is clear and acceptable. On the other hand, major revision is needed in order for this work to be published.
The problem that is addressed needs to be restated. In more details:
Initial geometry (why & external dimensions) Initial Mechanical analysis (which software) Mesh settings (i.e. uniform) Mesh Statistics (number of FE and nodes initial / final) FE Type Loading and boundary conditions (values & why) Results not visible in Figures 4, 5, 7 How does the FE model account for the dynamic effects with static structural analysis? Legends of Figures 4, 5, 9, 10, 12, 13, 15, 16 Contours not Clouds Legends of Figures 5, Deformation differs from Strain Topology optimization method Why use displacement in the objective function Elaborate on the constraint of 60% History of optimization iterations Final Mechanical analysis Mesh settings (uniform?) Mesh Statistics (number of FE and nodes initial / final) FE Type Loading and boundary conditions (values and why?) Experimental Figure 23 must be more detailed. The performance requirements must be better stated.
Author Response
Point 1: Initial geometry (why & external dimensions) Initial Mechanical analysis (which software) Mesh settings (i.e. uniform) Mesh Statistics (number of FE and nodes initial / final) FE Type Loading and boundary conditions (values & why) Results not visible in Figures 4, 5, 7 How does the FE model account for the dynamic effects with static structural analysis? Legends of Figures 4, 5, 9, 10, 12, 13, 15, 16 Contours not Clouds Legends of Figures 5, Deformation differs from Strain Topology optimization method Why use displacement in the objective function Elaborate on the constraint of 60% History of optimization iterations Final Mechanical analysis Mesh settings (uniform?) Mesh Statistics (number of FE and nodes initial / final) FE Type Loading and boundary conditions (values and why?) Experimental Figure 23 must be more detailed. The performance requirements must be better stated.
Response 1: For partial modification, Analyze suspension with dynamics software, and get the boundary conditions and loads. This is another part in report of formula car, I will use the results from report of formula car.
Yes ,use “contour” instead of “cloud”
Yes, use “deformation” instead of “strain” in Figures
This is first formula car, This year we will design the second formula car and get better results. Thanks for your advice.
Reviewer 3 Report
Dear Authors,
the required revisions and comments are listed in the attached PDF (wevj-622387-peer-review-v1 - final).
Although most of the comments are given in PDF, here are some additional remarks:
Methodology of your research is not clearly presented in the paper. A more detailed description of mathematical model, boundary conditions and loads would contribute to an easier understanding.
The paper will benefit from some good editing.

Author Response
For partial modification, please see attached new revision. ( Analyze suspension with dynamics software, and get the boundary conditions and loads.
Thanks!

Round 2
Reviewer 2 Report
Mesh settings (i.e. uniform)
Mesh Statistics (number of FE and nodes initial / final)
Results not visible in Figures 4, 5, 7
Author Response
please see attached revision

Reviewer 3 Report
Dear Authors,
Please, check line 30. The sentence you wright was copied from the review, but it in the review it was just a note to consider and check your original quote.
The paper will benefit from some good editing.
Author Response
please see attached revision
